

2       **Two-wavelength thermo-optical determination of Light Absorbing Carbon in**
         **atmospheric aerosols**

4       Dario Massabò[1,*], Alessandro Altomari[2], Virginia Vernocchi[1], Paolo Prati[1]

6       *1: Dept. of Physics, University of Genoa & INFN, Via Dodecaneso 33, 16146, Genova, Italy*

7       *2: Dept. of Physics, University of Genoa, Via Dodecaneso 33, 16146, Genova, Italy*

9                                       **Abstract**
Thermo-optical analysis is widely adopted for the quantitative determination of Total, TC,
Organic, OC, and Elemental, EC, Carbon in atmospheric aerosol sampled by suitable filters.
Nevertheless, the methodology suffers of several uncertainties and artefacts as the well-known
issue of charring affecting the OC-EC separation. In the standard approach, the effect of the
possible presence of Brown Carbon, BrC, in the sample is neglected. BrC is a fraction of OC,
usually produced by biomass burning with a thermic behaviour intermediate between OC and
EC. BrC is optically active: it shows an increasing absorbance when the wavelength moves to
the blue/UV region of the electromagnetic spectrum. Definitively, the thermo-optical
characterization of carbonaceous aerosol should be reconsidered to address the possible BrC
content in the sample under analysis.
We introduce here a modified Sunset Lab Inc. EC/OC Analyzer. Starting from a standard
commercial set-up, the unit has been modified at the Physics Department of the University of
Genoa (IT), making possible the alternative use of the standard laser diode at $\lambda = 635$ nm and
of a new laser diode at $\lambda = 405$ nm. In this way, the optical transmittance through the sample
can be monitored at both the wavelengths. Since at shorter wavelengths the BrC absorbance is
higher, a better sensitivity to this species is gained. The modified set-up also gives the
possibility to quantify the BrC concentration in the sample at both the wavelengths. The new
unit has been thoroughly tested, with both artificial and real-world samples: the first
experiment, in conjunction with the Multi Wavelength Absorbance Analyzer (MWAA,
Massabò et al., 2013 and 2015), resulted in the first direct determination of the BrC Mass
Absorption Coefficient (MAC) at $\lambda = 405$ nm: MAC $= 23 \pm 1$ m$^2$ g$^{-1}$.

***Keywords****: carbonaceous aerosol, brown carbon, thermo-optical analysis, mass absorption*
*coefficient*

---

*       Corresponding Author: massabo@ge.infn.it

## 1. Introduction

Light absorbing carbon (LAC) is the fraction of carbonaceous aerosol, which can absorb electromagnetic radiation in the visible or near-visible range (Pöschl, 2003; Bond and Bergstrom, 2006; Moosmüller et al., 2009; Ferrero et al., 2018). A wide literature investigates and characterizes the optical properties of the inorganic-refractory LAC fraction, usually referred as Black Carbon, BC, (e.g. Bond et al., 2013; and reference therein) which is strongly absorbing from UV to IR, with a weak dependence on wavelength (Bond and Bergstrom, 2006; Moosmüller et al., 2009). Much less studied and understood is the organic LAC, often labelled as Brown Carbon, which appears to be optically active at wavelengths shorter than 650 nm and with an increasing absorbance moving to the blue and ultraviolet (UV) range (Pöschl, 2003; Andreae and Gelencsér, 2006; Moosmüller et al., 2011; Laskin et al., 2015; Olson et al., 2015). BrC can therefore be considered as the "optically active" part of the OC dispersed in the atmosphere. When considered from a thermo-chemical point of view, BrC also shows a refractory behaviour since, in an inert atmosphere, it volatizes at temperatures greater than 400 °C only (Chow et al., 2015). A discussion on the sources of atmospheric LAC is outside the scope of the present work; we simply remind that it is produced mainly by biomass burning, even if in some cases also incomplete combustion of fossil fuels used in transport activities (i.e. terrestrial vehicles, ships and aircrafts) can generate this kind of compounds (Corbin et al, 2018). It is also worth to underline that carbonaceous aerosols impact on human health (Pope and Dockery, 2006; Chow et al., 2006; Mauderly and Chow, 2008), as well as on climate and environment (Bond and Sun, 2005; Highwood and Kinnersley, 2006; Chow et. al., 2010).

In the wider landscape of atmospheric carbonaceous aerosol, despite a worldwide diffused effort, the situation is not satisfactory and a standarized and conclusive approach is still missing. The quantitative determination of TC, OC and EC is often performed by a thermo-optical analysis (Birch and Cary, 1996; Watson et al., 2005; Hitzenberger et al., 2006) of aerosol samples collected on quartz fibre filters. However, thermo-optical analyses are affected by several issues and artefacts (Yang and Yu, 2002; Chow et al., 2004) and different laboratories/agencies adopt protocols which systematically result in discrepancies, particularly large in the EC quantification (Birch and Cary, 1996; Chow et al., 2007; Cavalli et al., 2010). A further issue arises when the effects of the possible presence of BrC in the sample are taken into account. So far, the monitoring of the sample transmittance during the thermal cycle, has been introduced to correct for the well know charring effect and the formation of pyrolytic carbon (Birch and Cary, 1996). This implies that BC is the sole absorbing compound at the



wavelength implemented in the thermo-optical analyser (for instance at λ = 635 nm, the
wavelength of the laser diode mounted in the extremely diffused Sunset Lab. Inc. EC/OC
analyzer). Basically, with a sizeable concentration of BrC in the sample, one of the key
assumptions of the thermo-optical methods fails and the EC/OC separation is even more
unstable (to not say that, by design, the BrC quantification is not possible). This issue was
preliminarily addressed by (Chen et al., 2015) by a multi-wavelength TOT/TOR instrument
(Thermal Spectral Analysis – TSA) and further investigated in (Massabò et al., 2016). In the
latter work, a method to correct the results of a standard Sunset analyzer and to retrieve the
BrC concentration in the sample was introduced. The achievement was possible thanks to a
synergy with the information provided by the Multi Wavelength Absorbance Analyzer,
MWAA, (Massabò et al., 2015) developed in the same laboratory. A further step towards BrC
quantification through the utilization of TSA was discussed in (Chow et al., 2018), where it
was proved that the use of 7-wavelengths in thermal/optical carbon analysis allows
contributions from biomass burning and secondary organic aerosols to be estimated. It is
worthy to note that the biomass burning contribution to PM concentration can be also estimated
by other methods such as Aerosol Mass Spectrometry, AMS (Daellenbach et al., 2016).
The MWAA approach allows the determination of the spectral dependence of the aerosol
absorption coefficient ($b_{abs}$) which can be generally described by the power-law relationship
$b_{abs}(\lambda) \sim \lambda^{-AAE}$, where the AAE is the Ångström Absorption Exponent. Several works reported
AAE values which depend on the aerosol chemical composition (Kirchstetter et al., 2004; Utry
et al., 2013) as well as its size and morphology (Lewis et al., 2008; Lack et al., 2012; Lack and
Langridge, 2013; Filep et al., 2013; Utry et al., 2014). Furthermore, the spectral dependence of
the aerosol has been exploited to identify different sources of carbonaceous aerosol (e.g.
Sandradewi et al., 2008; Favez et al., 2010; Lack and Langridge, 2013; Massabò et al., 2013
and 2015). In general, AAE values close to 1.0 have been found to be related to urban PM
where fossil fuels combustion is dominant, while higher AAE values, up to 2.5, have been
linked to carbonaceous aerosols produced by wood burning (Harrison et al., 2013; and
references therein) and therefore to the presence of BrC.
In the previous work by (Massabò et al., 2016) the effect of the BrC possibly contained in
the sample on the thermo-optical analysis was quantified and exploited to retrieve the BrC
concentration from the raw data provided by a standard Sunset Lab. Analyzer. This first step,
suggested to modify/upgrade a Sunset unit adding the possibility to use a second laser diode in
the blue range. This improves the sensitivity to the BrC and allows to check whether the BrC



quantification depends on the adopted wavelength. We finally followed this route and we here
introduce our modified Sunset Analyzer unit, the validation tests and the results of the first
campaign in which the new unit was deployed.

**2.   Materials and Methods**

*2.1 The 2-lambda SUNSET analyzer*

We have modified a commercial Thermal Optical Transmittance (TOT) instrument (Sunset

Lab Inc.). This equipment had been originally designed (Birch and Cary, 1996) with a red laser
diode ($\lambda = 635$ nm) to have the possibility to monitor and correct the well know problem of
the formation of pyrolytic carbon by charring (Birch and Cary, 1996; Bond and Bergstrom,
2006; Chow et al., 2007; Cavalli et al., 2010). The hypothesis under such choice was that the
OC is optically inactive at wavelengths greater than 600 nm and therefore the laser beam
attenuation is only due to the EC originally present or formed by charring in the sample under
analysis. Actually, even at this wavelength, BrC can affect the reliability of the OC/EC
separation and the standard methodology can be modified to quantify the BrC concentration
(Massabò et al., 2016). Nevertheless, at $\lambda = 635$ nm the BrC Mass Absorption Coefficient,
MAC(BrC), remains much smaller of the corresponding MAC(BC) and the modified
procedure could/should be implemented at shorter wavelengths to gain in sensitivity.

We have modified our SUNSET unit making possible the alternative use of the standard

laser diode at $\lambda = 635$ nm or of a World Star Technologies, 100 mW, laser diode at $\lambda = 405$
nm. This second laser diode can be mounted on the top of the SUNSET furnace by a homemade
adapter (see Figure 1) and easily exchanged with the native red diode. With the new laser
diode, the light detector placed at the bottom of the SUNSET furnace has to be changed too
and we selected a photodiode (PD) THORLABS FDS1010 coupled with a bandpass filter
THORLABS FBH405-10. The responsivity of the PD FDS1010 around $\lambda = 400$ nm is quite
low (about 50 mA W$^{-1}$) but the high power delivered by the laser diode results in signals with
an amplitude comparable to the values measured with the original SUNSET set-up.
Furthermore, the FBH405-10 filter cuts all the light background produced by the high
temperature of the SUNSET furnace, thus preserving the signal-to-noise ratio. Both laser and
PD can be exchanged in about 10 min and no particular attention is requested but the proper
alignment to maximize the PD output signal (i.e. the *transmittance* value displayed by the
SUNSET control software). We have to note that the original configuration of the SUNSET





instrument adopts a lock-in amplifier to improve the signal-to-noise ratio of the PD: we did not
have the possibility to manipulate the parameters of the lock-in amplifier and to tune it to the
new configuration.

### *2.2 Test of the new configuration*
The new set-up of the Sunset Analyzer was tested using both synthetic and real samples,
collected on quartz fibre filters. Synthetic samples were prepared starting with a 5% (volume)
solution of Aquadag, then nebulised by a Blaustein Atomizer (BLAM) and collected on quartz
fibre filters. Aquadag is the trade name of a water-based colloidal graphite coating (particle
diameters between 50 and 100 nm): these samples can therefore be considered to be composed
by EC/BC only. The samples were first sent to an optical characterization by the MWAA
instrument (Multi Wavelength Absorbance Analyzer, Massabò et al., 2015) which
demonstrated that the optical absorption of Aquadag is independent on the wavelength.
Actually, Aquadag particles tend to form conglomerates on the filters surface, with dimension
about double of the longer wavelength implemented in the MWAA (i.e. the 850 nm of the
infrared laser diode; Massabò et al., 2015). So, the comparison between the two Sunset set-ups
was made with samples having the same absorption properties. EC and TC quantifications
obtained at $\lambda = 635$ nm and $\lambda = 405$ nm resulted compatible adopting both the NIOSH5040 and
EUSAAR_2 protocol (Cavalli et al., 2010), as shown in Figure 2 for the whole set of synthetic
samples.
A second set of synthetic samples was prepared to mimic the behaviour of real-world aerosol
samples: a 3% (weight) solution of ammonium sulphate $(NH_4)_2SO_4$ in Aquadag was prepared
and nebulized with the BLAM. This way, a scattering compound is mixed to the absorbing
Aquadag spherules. The optical absorption measured with MWAA resulted independent on
wavelength with this second set of samples too. The results of the Sunset analysis with both
the red and blue laser set-up are shown in Figure 3. This second set of samples was analysed
through the EUSAAR_2 protocol only. A strong correlation between the TC and EC values
measured in red and blue light was obtained again with a slope close to unit.
A third and final test was performed using a set of daily PM10 samples collected by a low-
volume sampler (TCR - Tecora, Italy) on quartz fibre filters (Pall-2500 QAO-UP, 47 mm
diameter) in spring 2016 in the urban area of the city of Genoa (IT). A previous and long set
of similar campaigns addressed to PM10 characterization (e.g. Bove et al., 2014 and references
therein) in the same urban area could not identify sizeable contributes of biomass burning to
PM composition, in particular during spring and summer. Such situation was confirmed by the



determination of the Ångström exponent in the present samples by the MWAA. Actually, in
the set of twenty PM10 samples, the values of the Ångström exponent ranged between 0.9 and
1.2, this confirming that Black Carbon is the sole or totally dominant light absorbing
component in the local PM10 (Sandradewi et al., 2008; Harrison et al., 2013). Half of the
samples was then sent to the Sunset analysis by the NIOSH5040 protocol while the
EUSAAR_2 protocol was adopted for the remaining subset. The results are shown in Figure 4.
The EC concentration values measured with the standard and modified Sunset analyzer are
fully compatible when the NIOSH5040 protocol is adopted (basically, the split point position
in the Sunset thermogram does not change with the two laser diodes). Instead, EC values
determined by the EUSAAR_2 protocol resulted lower by about 30% when the blue laser diode
was mounted. This corresponds to a shift of the split point position, which moves right and thus
increases the amount of carbonaceous aerosol counted in the OC fraction. This effect is linked
to the well-known issue of the formation of pyrolytic carbon during the thermal cycle in the
inert atmosphere (i.e. in He). Several literature studies (e.g.: Cavalli et al., 2010; Panteliadis et
al., 2015) indicated that the charring is smaller at the higher temperatures reached during the
NIOSH thermal protocol. On the other way, standard thermo-optical analyses of urban PM
samples often give higher EC values (up to 50%) when performed following the EUSAAR_2
instead of higher-temperature protocols (Subramanian et al., 2006; Zhi et al., 2008; Piazzalunga
et al., 2011; Karanasiou et al., 2015; Panteliadis et al., 2015). Furthermore, as by-product of
previous PM10 studies in the urban area of Genoa by a standard Sunset unit, we could observe
a systematic and very reproducible 40% discrepancy between EC values determined in the
same samples by EUSAAR_2 and NIOSH5040 protocols (with EC:EUSAAR > EC:NIOSH).
Therefore, the thermo-optical analysis in blue light seems to be more sensitive to the charring
formation during the EUSAAR_2 protocol and thus possibly more reliable in the EC/OC
separation.

**3.   First field campaign and results**

The modified Sunset set-up was used for the first time, in conjunction with the MWAA
instrument and apportionment methodology (Massabò et al., 2015), to retrieve the MAC (Mass
Absorption Coefficient) of Brown Carbon at the two wavelengths of $\lambda = 635$ nm and $\lambda = 405$
nm, in a set of samples collected wintertime in a mountain site.

*3.1 Samples collection*



Aerosol samples were collected in a small village (Propata, 44°33'52.93''N, 9°11'05.57''E,
970 m a.s.l.) situated in the Ligurian Apennines, Italy. Three different sets of PM10 aerosol
samples were collected by a low-volume sampler (38.3 l min⁻¹ by TCR Tecora): the first and
the third sets had filter change set every 24h while the second set was sampled on a 48h-basis.
In total, 41 (14+13+14) PM10 samples were collected on quartz-fibre filters (Pall, 2500QAO-
UP, 47 mm diameter), between February 2nd and April 19th, 2018. Before the sampling, the
filters were baked at T = 700°C for 2 hours to remove possible internal contamination. Field
blank filters were used to monitor possible contaminations during the sampling phase. Wood
burning is one of the PM sources around the sampling site, especially during the cold season,
as it is used for both domestic heating and cooking purposes.

### 3.2 Laboratory analyses
All the samples were weighed before and after sampling in an air-conditioned room (T = 20
± 1 °C; R.H. = 50 % ± 5%), after 48h conditioning. The gravimetric determination of the PM
mass was performed using an analytical microbalance (precision: 1 μg) which was operated
inside the conditioned room; electrostatic effects were avoided by the use of a de-ionizing gun.
After weighing, samples were first optically analyzed by MWAA to retrieve the absorption
coefficient ($b_{abs}$) of PM at five different wavelengths. The EC and OC determination was
performed adopting the EUSAAR_2 protocol (Cavalli et al., 2010) with both laser diodes at λ
= 635 nm and at λ = 405 nm (two different punches were extracted from each filter sample).
Finally, the remaining portion of the same quartz-fibre filters underwent a chemical
determination of the Levoglucosan (1,6-Anhydro-beta-glucopyranose) concentration by High
Performance Anion Exchange Chromatography coupled with Pulsed Amperometric Detection
(Piazzalunga et al., 2010). As well known in literature, this sugar is one of the typical marker
of biomass burning (Vassura et al., 2014).

### 3.3 Optical apportionment
The MWAA analysis provided the raw data to measure the spectral dependence of the
aerosol absorption coefficient ($b_{abs}$) which can be generally described by the power-law
relationship $b_{abs}(\lambda) \sim \lambda^{-AAE}$ where AAE is the Ångström Absorption Exponent.
The time series of the resulting AAE values is shown in Figure 5: they range between
1.05 and 1.96 with a mean value of 1.55 ± 0.21. This figure indicates a substantial presence of
wood burning in the sampling area. In (Massabò et al., 2015 and Bernardoni et al, 2017), an



optical apportionment model (the "MWAA model") based on the measurement of $b_{abs}$ at five
wavelengths had been introduced to obtain directly the BrC AAE ($\alpha_{BrC}$) and the BrC absorption
coefficient ($b_{abs}^{BrC}$) at each measured wavelength. It is worthy to note that, at the basis of the
MWAA model, there is the assumption that BrC is produced by wood combustion only (see
§4 in Massabò et al., 2015; Zheng et al., 2013). In Figure 5, we report the optical apportionment
at $\lambda = 635$ nm and at $\lambda = 405$ nm i.e. at the wavelength of the two laser diodes used in our
modified Sunset instrument. At $\lambda = 635$ nm, light absorption resulted mainly due to BC from
both fossil fuel (FF) and biomass burning (WB) and the $b_{abs}^{BrC}$ average value is 15% of total
$b_{abs}$, with the notable exception of some days in which it reached values of ~ 30%, in
correspondence of AAE > 1.9. Instead, at $\lambda = 405$ nm, the BrC contribute to light absorption
rises up to 33% (average percentage of total $b_{abs}$), with a maximum value of 51%, again when
$AAE_{exp} > 1.9$. The time series of $b_{abs}^{BrC}$ values at both the wavelengths turned out to be well
correlated ($R^2 = 0.71$) with the Levoglucosan (*Levo*, in the following) concentration values, as
reported in Figure 6. The slope of the correlation curve increases by a factor 5.8 when moving
from the red to the blue light.
The average $\alpha_{BrC}$ value turned out to be $\alpha_{BrC} = 3.9 \pm 0.1$, in very good agreement with a
previous value ($\alpha_{BrC} = 3.8 \pm 0.2$) obtained in the same site and with the same approach
(Massabò et al., 2016). The result is also in agreement with other literature works (Yang et al.,
2009; Massabò et al., 2015; Chen et al., 2015).

### 3.4 Brown Carbon MAC

The methodology to extract the MAC value for BrC by the coupled used of MWAA and
Thermo-Optical Analysis has been introduced in a previous work (Massabò et al., 2016). In
that case, a standard (i.e.: with a red laser diode only) Sunset unit was used. The entire
procedure is described in details in (Massabò et al., 2016), here we briefly summarize the main
steps:
a) The fraction of light attenuation due to the BrC is first calculated in each sample with
the MWAA raw data.
b) The empirical relationship between the light attenuation through the sample, observed
in the MWAA and Sunset set-ups is then determined. We remind that in the Sunset set-
up, the light attenuation is continuously recorded during the analysis; the value
characteristic of each blank filter can be retrieved when all the light absorbing PM has
been volatized (i.e. at the end of the thermal protocol).



c) The fraction of light attenuation due to the BrC in the sample is therefore calculated for the Sunset set-up and the initial transmittance value is corrected to estimate the attenuation value that it would have been found if BrC were not present in the filter sample.

d) A new split-point position is then determined taking into account the corrected value of the initial transmittance.

e) The OC and EC values determined with the standard and corrected split-point positions are then compared and the difference ($OC_{cor} - OC_{std} = EC_{std} - EC_{cor}$) is operatively assumed to be equal to the BrC in the sample. The corresponding BrC atmospheric concentration is finally calculated.

f) The correlation between the values of $b_{abs}^{BrC}$, provided by the MWAA analysis (see section 3.3) and BrC concentration, is studied to determine the MAC value.

In the present experiment, the procedure was adopted to analyse the thermograms produced with both the red and the blue laser diode mounted in the Sunset unit: the results are summarized in Figure 7. Despite a rather high noise in the data, the MAC(BrC) value at the two wavelengths can be determined and it turns out to be MAC(BrC) = $9.8 \pm 0.4$ m$^2$ g$^{-1}$ and $23 \pm 1$ m$^2$ g$^{-1}$, respectively at $\lambda = 635$ and 405 nm. This result deserve some comments:

- The MAC value at $\lambda = 635$ nm differs for less than $3\sigma$ from the result reported in (Massabò et al., 2016) and obtained in the same site and in a similar season (i.e. November 2015 to January 2016; MAC = $7.0 \pm 0.4$ m$^2$ g$^{-1}$). Since differences in the type of wood burnt in the past and present campaign cannot be excluded, the two values can be considered to be in fair agreement.

- No comparison with previous or other literature values is possible for the MAC value at $\lambda = 405$ nm, given the substantial differences in adopted definitions and methodologies (Yang et al., 2009; Feng et al., 2013; Chen and Bond, 2010). However, the increase by a factor 2.3 with respect to the MAC at $\lambda = 635$ nm follows the expected behaviour.

- Under the assumption that the sole source of BrC is biomass burning, the MAC values can be referred to the total concentration of organic carbon (i.e. including the part not optically active) produced by biomass burning. Adopting with the present data set the optical OC apportionment methodology reported in (Massabò et al., 2015), the BrC values determined at $\lambda = 635$ nm turn out to be about 4% of the OC produced by wood




combustion, $OC_{WB}$, and consequently $MAC(OC_{WB,} @635nm) = 0.39 \pm 0.06$ m$^2$ g$^{-1}$.
When the analysis is performed at $\lambda = 405$ nm, BrC results to be about 10% of $OC_{WB}$
and $MAC(OC_{WB,} @405nm) = 2.3 \pm 0.2$ m$^2$ g$^{-1}$. Previous literature works (Feng et al.,
2013; Laskin et al., 2015; and references therein) report MAC values of BrC and/or
related OC ranging in a quite large interval.
• The ratio between BrC and Levo concentration values results to be BrC:Levo = 0.19 ±
0.02 and 0.42 ± 0.06, respectively when considering the BrC concentration determined
by MWAA+Sunset at $\lambda = 635$ and 405 nm. In other words, the operative procedure,
introduced in (Massabò et al., 2016), results in different BrC concentration values
according to the considered/used wavelength. This fact can be interpreted in different
ways: while the analytical sensitivity is higher at $\lambda = 405$ nm and the corresponding
BrC values could be considered to be more firm, the category of compounds collected
under the label "Brow Carbon" could be itself "wavelength dependent". The latter
would imply that the BrC concentration cannot be defined separately from the
wavelength and that its meaning is even more "operative" of the more widespread OC
and EC fractions. As a matter of fact, while the $b_{abs}{}^{BrC}$ values discussed in section 3.3,
increase by a factor 5.8 moving from $\lambda = 635$ nm to $\lambda = 405$ nm, the corresponding
variation of the MAC(BrC) values is by a factor 2.3 only. This is because the BrC
concentration determined at $\lambda = 405$ nm doubles the value measured at $\lambda = 635$ nm.
The purposes and the limits of the present study prevent any firm conclusion on the
alternative explanation: BrC definition is wavelength dependent or the analysis in red
light is not sensitive enough.
• When considering the $OC_{WB}$:Levo concentration ratio, the MWAA analysis at $\lambda = 635$
and $\lambda = 405$ nm give well compatible results, with a mean value of $OC_{WB}$:Levo = 4.5

± 0.5.

## 4. Conclusions

We introduced a modified version of a commercial Sunset Lab. Inc. OC/EC Analyzer. We
upgraded the set-up of a standard unit making possible the alternative use of a red ($\lambda = 635$
nm) or blue ($\lambda= 405$ nm) laser diode to monitor the light transmittance through the sample
during the thermal cycle. The analytical performance of the new set-up has been tested both
with artificial and real-world samples.




The new Sunset unit was used for the first time to analyze a set of samples collected mostly
wintertime in a mountain site of the Italian Apennines. Exploiting the synergic information
provided by the Multi Wavelength Absorbance Analyzer, MWAA (Massabò et al., 2015) and
adopting the procedure to retrieve the Brown Carbon concentration directly from the Sunset
thermograms (Massabò et al., 2016), we could measure the MAC(BrC) at the two wavelengths.
The result at $\lambda = 635$ nm (MAC = $9.8 \pm 0.4$ m$^2$ g$^{-1}$) is in fair agreement with a previous study
performed in the same site in winter 2015-2016. At our knowledge, the result at $\lambda = 405$ nm,
MAC = $23 \pm 1$ m$^2$ g$^{-1}$ is the sole direct observation at this wavelength.
In our analysis, the ratio between the BrC and the Levo concentration values depends on the
wavelength adopted during the thermo-optical analysis. This behaviour could be due to a better
accuracy of the results in blue-light, more sensitive at the BrC, or because the definition of BrC
itself has to be considered wavelength-dependent. The present results do not allow any
conclusive statement on this issue: actually, the label "Brown Carbon" as well as the widely
used "Organic and Elemental Carbon" comes from an operative definition not without
ambiguity.

**6. Acknowledgements**

This work has been partially financed by the National Institute of Nuclear Physics
(INFN) in the frame of the TRACCIA experiments.

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



**FIGURE CAPTIONS**

**Figure 1:** The new $\lambda = 405$ nm laser diode mounted by a steel adapter on the SUNSET furnace (top) and the comparison with the standard $\lambda = 635$ nm laser diode implemented by the manufacturer (bottom).

**Figure 2**: Quantification of TC (primary axis) and EC (secondary axis) at $\lambda = 635$ nm (red) and $\lambda = 405$ nm (blue) for the set of synthetic Aquadag samples. Top: NIOSH5040 protocol, bottom: EUSAAR_2 protocol.

**Figure 3**: Quantification of TC (primary axis) and EC (secondary axis) at $\lambda = 635$ nm (red) and $\lambda = 405$ nm (blue) for the set of synthetic Aquadag + Ammonium Sulphate samples by the EUSAAR_2 protocol.

**Figure 4**: EC concentration measured in two sub-sets of PM10 samples collected in consecutive days in the urban area of Genoa in late spring 2016. Values determined with the Sunset analyzer equipped with blue and red laser diodes, are compared.

**Figure 5**: Primary axis: Optical apportionment of the aerosol absorption coefficient ($b_{abs}$) @ $\lambda = 635$ nm (top) and $\lambda = 405$ nm (bottom). Secondary axis: experimental AAE values obtained by fitting the measured $b_{abs}$ values with a power-law relationship $b_{abs}(\lambda) \sim \lambda^{-AAE}$. WW and FF stand for Fossil Fuel and Wood Burning, respectively.

**Figure 6**: Aerosol absorption coefficient apportioned to Brown Carbon ($b_{abs}^{BrC}$) @635 nm (top) and @405 nm (bottom) vs. levoglucosan concentration.

**Figure 7**: Comparison between the aerosol absorption coefficient apportioned to Brown Carbon vs. the resulting operative BrC concentration values @635 nm (top) and @405 nm (bottom).



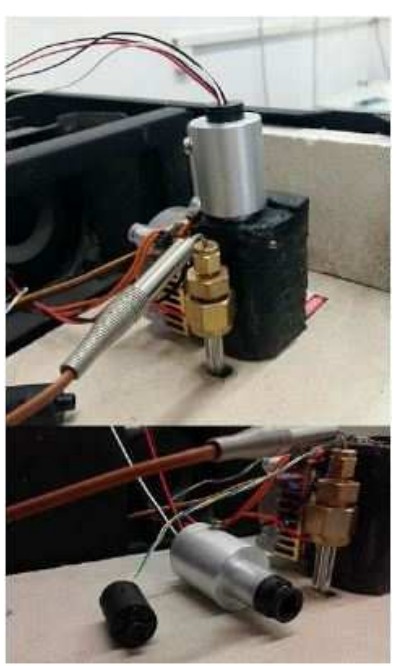


Figure 1






Figure 2




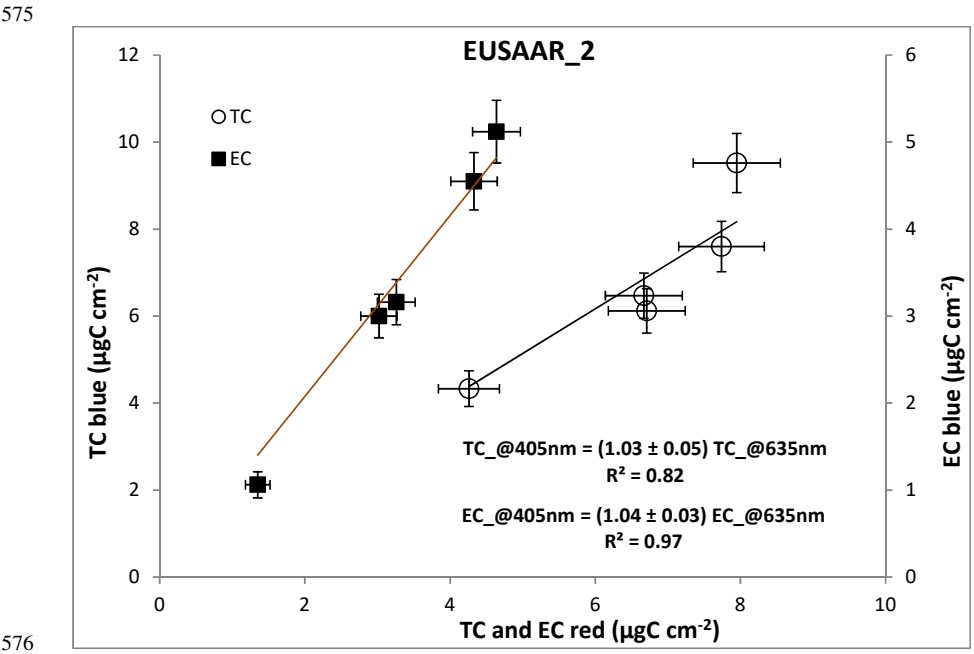

Figure 3

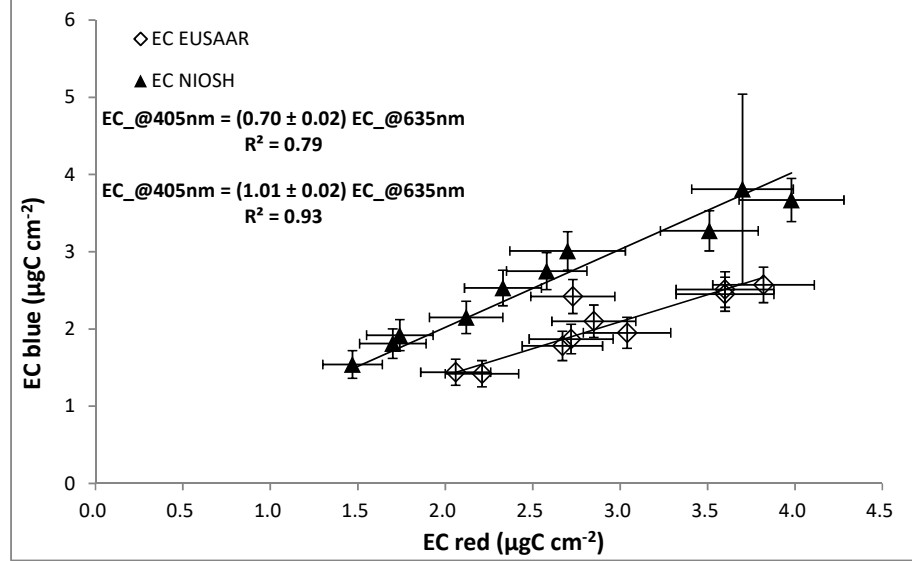

Figure 4




Figure 5






Figure 6






Figure 7