# Peer review of "1. Introduction"

_Atmospheric Measurement Techniques, 2019_

## Referee Comment (RC1) · Anonymous Referee #2 · 20 Feb 2019

The aim of the paper should be better stressed in the abstract. Furthermore from the abstract it is not clear why the authors decided to use a different laser radiation. It has to be specified. The text could be simplified in many points. For example at line 64: aerosol samples collected on filters made of refractory material (i.e. quartz fibre filters), should be: . . . made of quartz fibre filters (in fact there aren't other possibility). From line 69 to line 72 the text could be simplified since how TOT method operates is well known. At line 85 of the introduction it should be mentioned that biomass burning contribution to PM could be also estimated by other methods such as AMS (aerosol mass spectrometry) which starting from mass spectral data is suitable for identification of biomass burning source. At this purpose I suggest to include the following reference:

[Figure]

Daellenbach, K.R., Bozzetti, C., Křepelová, A., Canonaco, F., Wolf, R., Zotter, P., Fermo, P., Crippa, M., Slowik, J.G., Sosedova, Y., Zhang, Y., Huang, R.-J., Poulain, L., Szidat, S., Baltensperger, U., El Haddad, I., Prévôt, A.S.H. Characterization and source apportionment of organic aerosol using offline aerosol mass spectrometry (2016) Atmospheric Measurement Techniques, 9 (1), pp. 23-39.

Somewhere in the introduction it should be mentioned that levoglucosan is the marker for BB. At this purpose together with the reference Piazzalunga 2010, I suggest to include for example the following reference:

Vassura, I., Venturini, E., Marchetti, S., Piazzalunga, A., Bernardi, E., Fermo, P., Passarini, F. Markers and influence of open biomass burning on atmospheric particulate size and composition during a major bonfire event (2014) Atmospheric Environment, 82, pp. 218-225.

Please also note the supplement to this comment:
https://www.atmos-meas-tech-discuss.net/amt-2019-5/amt-2019-5-RC1-supplement.pdf
* * *

---

## Referee Comment (RC2) · Anonymous Referee #3 · 14 Mar 2019

**Review for amt-2019-5 "Two-wavelength thermo-optical determination of Light Absorbing Carbon in atmospheric aerosols" by Massabò et. al.**

The manuscript reported a modified carbon analyzer with dual-wavelength configuration for the determination of Brown Carbon. Multiwavelength thermal-optical analysis had been reported using the DRI carbon analyzer (Chen et al., 2015; Chow et al., 2015; Chow et al., 2018), but multiwavelength applications on the Sunset analyzer remain limited (Hadley et al., 2008). In that sense, this study has the merit from the instrumental perspective. However, part of the data analysis suffered from overinterpretation, thus, revisions are needed.

**General comments:**

1) How this study can be beneficial to the carbonaceous aerosol research community? From the instrumental perspective, there is already a multiwavelength carbon analyzer that is commercially available (DRI2015), as noted by the authors. The modification described in this study might not be easy to be adopted and implemented by other research groups. The authors need to elaborate how this setup can be implemented by other researchers.

2) Introduction. Beside primary BrC from biomass burning, the secondarily formed BrC should be mentioned.

3) The current modification only allows one laser to be used at each time, that means all samples need to be analyzed twice. As noted by the authors, the change of laser and PD require alignment to optimize the laser signal. Since laser and PD change would introduce further uncertainties into the OC/EC analysis, this point should be mentioned. Did the authors quantify the uncertainties in OC and EC determination that introduced by the change of laser and PD? For example, what's the standard deviations of OC and EC from multiple analysis for the same sample (identical laser and PD with mount-unmount cycles scenario vs. no laser and PD change scenario)? The authors are encouraged to provide a estimation of uncertainty introduced.

4) The MAC$_{BrC}$ reported in this study (9.8 m$^2$g$^{-1}$ @635 nm and 23 m$^2$g$^{-1}$ @405 nm) seems to be one magnitude higher than the literatures values. An example is shown below. The following table was adopted from (Updyke et al., 2012). The author argued the difference is due to the operative defined BrC mass used in this study. It should be noted that literature studies applied different technical approaches for MAC$_{BrC}$ determination as well, but most studies reported a MAC$_{BrC}$<1 m$^2$g$^{-1}$. The author should explain why their results are significantly different from previous studies.

Values of MAC in m$^2$ g$^{-1}$ from representative field and laboratory studies of organic aerosols measured at or near 500 nm. The cited MAC values explicitly remove contributions from black carbon. With the exception of this work, all data cited in this table likely correspond to primary sources of brown carbon.

| Sample (campaign) | λ (nm) | MAC (m$^2$ g$^{-1}$) | Reference |
|---|---|---|---|
| Brown carbon produced by aging SOA with 100 ppb NH$_3$ (lab) | 500 | 0.001−0.1 | This work |
| "Tar balls" from smoldering combustion of wood; brown carbon contribution (lab) | 532 | 0.01−0.07 (calculated from $k$ = 0.0005−0.003) | (Chakrabarty et al., 2010) |
| HUmic-LIke Substances (HULIS) extracted from filter samples from various sites in Europe | 532 | 0.07−1 (calculated from $k$ = 0.003−0.05) | (Dinar et al., 2008) |
| Methanol extracts from wood combustion particles (lab) | 500 | 0.1−0.5 (estimated from the graphs) | (Chen and Bond, 2010) |
| Refractory organic carbon from biomass burning in North America (INTEX/ICARTT) | 470 | 0.6 | (Clarke et al., 2007) |
| | 530 | 0.1 | |
| "Tar balls" in North America (YACS) | 632 | 0.4 (calculated from reported $k$ = 0.02) | (Hand et al., 2005) |
| Brown carbon in pollution plumes from Asia (CAPMEX) | 532 | 0−1 | (Flowers et al., 2010) |
| Brown carbon in particles collected in Asia (EAST-AIRE) | 520 | 0.6 | (Yang et al., 2009) |
| Acetone extracts form biomass burning aerosols in Africa (SAFARI 2010) | 500 | 0.9 | (Kirchstetter et al., 2004) |
| Amorphous carbon spheres from biomass burning (ACE Asia) | 550 | 4 | (Alexander et al., 2008) |

5) Line 277. Regarding the BrC mass determination using the method reported in Massabò et al. (2016), did the author considered laser-temperature correction (Jung et al., 2011) ? Seen from Fig 5 in Massabò et al. (2016), the laser signal keep increasing during the $CH_4$ stage, implying that laser-temperature correction was likely not performed. If that's case, the BrC mass should be re-calculated. In addition, even if the laser-temperature correction is accounted, the laser uncertainty is simply too high for BrC mass determination. Please specify the limit of quantification (LOQ) for OC in the OC/EC analysis. The reviewer feels that $LOQ_{OC}$ would be likely very close to the level of BrC reported in this study ($0.005 - 0.14$ µgC m$^{-3}$). If so, BrC reported using this approach is overinterpretation of the data.

6) The BrC determination approach described in Massabò et al. (2016) lacks physical meanings. The OC/EC split by the laser signal in the thermal optical analysis depends on two assumptions: (i) pyrolyzed organic carbon evolved before native EC during the oxygen stage. (ii) pyrolyzed organic carbon and native EC have the same MAC. However, both of these assumptions had been proved invalid (Yang and Yu, 2002; Yu et al., 2002; Subramanian et al., 2006). The approach that author used is a paradox: On one hand the authors report a $MAC_{BrC}$ that is larger than $MAC_{EC}$. On the other hand, the laser correction process itself is based on the assumption that $MAC_{BrC}=MAC_{EC}=MAC_{POC}$. In that sense, the carbon fraction corresponding to the different laser split time cannot be considered as BrC mass.

7) The authors are encouraged to check the $b_{abs,BCff}$ vs. levo scatter plot. If the $R^2(b_{abs,BCff}$ vs. levo) is significantly lower than the $R^2(b_{abs,BrC}$ vs. levo), that would be a useful evidence to confirm a successful split of $b_{abs}$ into BrC, $BC_{WB}$ and $BC_{ff}$.

**Technical comments:**

1) The figure quality needs to be improved. For example, for comparison of the same quantity/parameter, the X and Y range should be the same and the aspect ratio of the plot should be 1:1.

2) Figure 1. Please label the laser wavelength on the photo directly for easy reference.

3) Figure 2-7. The font size is too small for the text in these figures. Please adjust accordingly.

4) Figure 5 caption. "WW and FF stand for Fossil Fuel and Wood Burning, respectively." Should be "FF and WB stand for Fossil Fuel and Wood Burning, respectively"

5) Line 245. "and biomass burning (WB)" should be wood burning?

**References**

Chen, L. W. A., Chow, J. C., Wang, X. L., Robles, J. A., Sumlin, B. J., Lowenthal, D. H., Zimmermann, R., and Watson, J. G.: Multi-wavelength optical measurement to enhance thermal/optical analysis for carbonaceous aerosol, Atmos. Meas. Tech., 8, 451-461, doi: 10.5194/amt-8-451-2015, 2015.

Chow, J. C., Wang, X., Sumlin, B. J., Gronstal, S. B., Chen, L. W. A., Trimble, D. L., Kohl, S. D., Mayorga, S. R., Riggio, G., Hurbain, P. R., Johnson, M., Zimmermann, R., and Watson, J. G.: Optical Calibration and Equivalence of a Multiwavelength Thermal/Optical Carbon Analyzer, Aerosol. Air. Qual. Res., 15, 1145-1159, doi: 10.4209/aaqr.2015.02.0106, 2015.

Chow, J. C., Watson, J. G., Green, M. C., Wang, X., Chen, L. W. A., Trimble, D. L., Cropper, P. M., Kohl, S. D., and Gronstal, S. B.: Separation of brown carbon from black carbon for IMPROVE and Chemical Speciation Network PM2.5 samples, J. Air Waste Manage. Assoc., 68, 494-510, doi: 10.1080/10962247.2018.1426653, 2018.

Hadley, O. L., Corrigan, C. E., and Kirchstetter, T. W.: Modified Thermal-Optical Analysis Using Spectral Absorption Selectivity To Distinguish Black Carbon from Pyrolized Organic Carbon, Environ. Sci. Technol., 42, 8459-8464, doi: 10.1021/Es800448n, 2008.

Jung, J., Kim, Y. J., Lee, K. Y., Kawamura, K., Hu, M., and Kondo, Y.: The effects of accumulated refractory particles and the peak inert mode temperature on semi-continuous organic carbon and elemental carbon measurements during the CAREBeijing 2006 campaign, Atmos. Environ., 45, 7192-7200, doi: 10.1016/j.atmosenv.2011.09.003, 2011.

Massabò, D., Caponi, L., Bove, M. C., and Prati, P.: Brown carbon and thermal–optical analysis: A correction based on optical multi-wavelength apportionment of atmospheric aerosols, Atmos. Environ., 125, 119-125, doi: 10.1016/j.atmosenv.2015.11.011, 2016.

Subramanian, R., Khlystov, A. Y., and Robinson, A. L.: Effect of peak inert-mode temperature on elemental carbon measured using thermal-optical analysis, Aerosol. Sci. Technol., 40, 763-780, doi: 10.1080/02786820600714403, 2006.

Updyke, K. M., Nguyen, T. B., and Nizkorodov, S. A.: Formation of brown carbon via reactions of ammonia with secondary organic aerosols from biogenic and anthropogenic precursors, Atmos. Environ., 63, 22-31, doi: 10.1016/j.atmosenv.2012.09.012, 2012.

Yang, H. and Yu, J. Z.: Uncertainties in charring correction in the analysis of elemental and organic carbon in atmospheric particles by thermal/optical methods, Environ. Sci. Technol., 36, 5199-5204, doi: 10.1021/Es025672z, 2002.

Yu, J. Z., Xu, J. H., and Yang, H.: Charring characteristics of atmospheric organic particulate matter in thermal analysis, Environ. Sci. Technol., 36, 754-761, doi: 10.1021/Es015540q, 2002.

---

## Referee Comment (RC3) · Anonymous Referee #1 · 19 Mar 2019

The paper describes the application of a modified Sunset Lab Inc. EC/OC analyzer with a two-wavelength set-up for analysis of ambient aerosol samples. It strongly relates to earlier work of the author which is described detailed in previous publications, nevertheless it extends to additional findings. The results derived from the comparison of two temperature protocols, NIOSH5040 and EUSAAR2, for the 405 nm wavelength as well as the reported MAC values are a valuable addition to the literature. However, the paper lacks in structure and suffers in use of proper scientific English which limits its potential. It is recommended that the text is reviewed with focus on syntax and vocabulary. Particular attention should be given in the first paragraph of the abstract and the first two paragraphs of the introduction. Further:

Sentence starting in line 115: "The hypothesis under such choice..." should be rephrased.

The terms "real samples", "real-world samples" and "real-world aerosol samples" are used throughout the text. The use of one term is recommended.

"$\lambda=$", "$@\lambda$" and "$@\lambda=$" are used to state wavelengths. Please consider using one form ($\lambda=$) for consistency.

Sentence starting in line 59 should be revised. "Standarized" should be replaced by standardized. Since 2017, when EN16909 was published, there is uniformity in OC/EC analysis methodology, at least for EU.

Paragraph starting in line 157: Any specific reason why this subset was analyzed with EUSAAR2 only?

Paragraph starting in line 165: PM10 samples are known to add complexity in OC/EC analysis due to minerals, refractory material and oxides present in coarse fraction. Did you consider sampling/analysis of PM2.5 samples and have you noticed any of the above interferences?

Line 174: It sounds like two different subsets were created, one for analysis with EU-SAAR2 and one for NIOSH5040. If that is the case, why was that choice made instead of all samples being analyzed with both protocols?

Line 193: It is not clear to me why the discrepancy between EUSAAR2 and NIOSH5040 is mainly driven by charring. In a sense more pyrolytic carbon would result in a later split point and less EC reported. Further, since the blue laser diode resulted in later split points for EUSAAR2, wouldn't that rate it as less sensitive to charring instead of more, as mentioned in the text?

Is it possible to include a figure and/or representative thermograms that illustrate the consistent 40% discrepancy between EUSAAAR2 and NIOSH5040?

Line 267 and elsewhere in the text: The term "Sunset set-ups" could be altered to a more descriptive term.

Line 289: What would be the value of $1\sigma$? It seems that the difference between the two MAC values reported is substantially greater than the reported uncertainty.

Line 316: "Brow carbon"

Line 338: This sentence could be rephrased for easier comprehension.

Line 348: Same stands for this sentence.

Figure 4: It is not clear which relationship applies to which trendline.

Figure 6: It seems that 2 separate subgroups are formed, one equal and above the trendline and one below the trendline. Are those related to the specific sampling strategy or to any other parameter?

Please also note the supplement to this comment:
https://www.atmos-meas-tech-discuss.net/amt-2019-5/amt-2019-5-RC3-supplement.pdf

---

## Author Comment (AC1) · 17 Apr 2019

*Answers to RC1: 'general comments', Anonymous Referee #2*

*We strongly acknowledge the Referee for the valuable and precious comments and suggestions. In the following, a point-by-point reply to all the comments is given.*

**The aim of the paper should be better stressed in the abstract**. Furthermore from the abstract it is not clear why the authors decided to use a different laser radiation. **It has to be specified**.

*Following the Referee's advice we added in the abstract few sentences to better stress the aim of the paper and why we decided to use a different laser wavelength.*

The text could be simplified in many points. For example at **line 64**: aerosol samples collected on filters made of refractory material (i.e. quartz fibre filters), should be: … made of quartz fibre filters (in fact there aren't other possibility).

*Done*

**From line 69 to line 72** the text could be simplified since how TOT method operates is well known.

*In our opinion, this very short paragraph could be useful to a reader not familiar with TOT measurements. So we prefer to keep it in the text.*

At **line 85** of the introduction it should be mentioned that biomass burning contribution to PM could be also estimated by other methods such as AMS (aerosol mass spectrometry) which starting from mass spectral data is suitable for identification of biomass burning source. At this purpose I suggest to include the following reference:

Daellenbach, K.R., Bozzetti, C., Křepelová, A., Canonaco, F., Wolf, R., Zotter, P., Fermo, P., Crippa, M., Slowik, J.G., Sosedova, Y., Zhang, Y., Huang, R.-J., Poulain, L., Szidat, S., Baltensperger, U., El Haddad, I., Prévôt, A.S.H. Characterization and source apportionment of organic aerosol using offline aerosol mass spectrometry (2016) Atmospheric Measurement Techniques, 9 (1), pp. 23-39.

A short statement and the suggested reference have been added in the text

Somewhere in the introduction, it **should be mentioned that levoglucosan is the marker for BB**. At this purpose together with the reference Piazzalunga 2010, I suggest to include for example the following reference:

Vassura, I., Venturini, E., Marchetti, S., Piazzalunga, A., Bernardi, E., Fermo, P., Passarini, F. Markers and influence of open biomass burning on atmospheric particulate size and composition during a major bonfire event (2014) Atmospheric Environment, 82, pp. 218-225.

A short statement and the suggested reference have been added in the last part of section 3.2

---

## Author Comment (AC2) · 17 Apr 2019

**Review for amt-2019-5 "Two-wavelength thermo-optical determination of Light Absorbing Carbon in atmospheric aerosols" by Massabò et. al.**

*We strongly acknowledge the Referee for the valuable and precious comments and suggestions.*

The manuscript reported a modified carbon analyzer with dual-wavelength configuration for the determination of Brown Carbon. Multiwavelength thermal-optical analysis had been reported using the DRI carbon analyzer (Chen et al., 2015; Chow et al., 2015; Chow et al., 2018), but multiwavelength applications on the Sunset analyzer remain limited (Hadley et al., 2008). In that sense, this study has the merit from the instrumental perspective. However, part of the data analysis suffered from overinterpretation, thus, revisions are needed.

*In the following, a point-by-point reply to all the comments is given.*

**General comments:**

1) How this study can be beneficial to the carbonaceous aerosol research community? From the instrumental perspective, there is already a multiwavelength carbon analyzer that is commercially available (DRI2015), as noted by the authors. The modification described in this study might not be easy to be adopted and implemented by other research groups. The authors need to elaborate how this setup can be implemented by other researchers.

   *The Referee is right, since a multi-wavelength carbon analyzer is commercially available, but, at least at the moment, it is not so widespread. Especially in Europe, the one wavelength Sunset EC/OC analyzer unit is by far the most common instrument for this kind of quantification. We think that the possibility to upgrade these "old" units to make possible 2-$\lambda$ measurements is a good chance to have more information on EC/OC separation and to study the effect of BrC (but not only) on TOT analysis. Moreover, the upgrade is easy and cheap since just a blue LED and a photodiode/bandpass filter system are needed. The basic information to implement this upgrade are detailed in the paper; for sure we are available to help anybody to carry out this operation but more technical issues should be given outside the present paper.*

2) Introduction. Beside primary BrC from biomass burning, the secondarily formed BrC should be mentioned.

   *The Referee is right, we forgot to mention the secondary formation of BrC. It has been inserted in the text.*

3) The current modification only allows one laser to be used at each time, that means all samples need to be analyzed twice. As noted by the authors, the change of laser and PD require alignment to optimize the laser signal. Since laser and PD change would introduce further uncertainties into the OC/EC analysis, this point should be mentioned. Did the authors quantify the uncertainties in OC and EC determination that introduced by the change of laser and PD? For example, what's the standard deviations of OC and EC from multiple analysis for the same sample (identical laser and PD with mount-unmount cycles scenario vs. no laser and

PD change scenario)? The authors are encouraged to provide a estimation of uncertainty introduced.

*We have conducted several repetitions in order to evaluate uncertainties coming from the laser switching procedure. Uncertainties turned out to be of the same order of magnitude of the typical value characteristic of TOT measurements, i.e. about ± 10%, so we did not discuss uncertainties in the text. Anyway, the Referee is right since the procedure is simple but in principle effects due to different laser alignments cannot be excluded. For this reason, we do not change the lasers filter by filter, but we first perform the analysis of the whole samples batch with the red laser and then we repeat the analysis with the blue diode. In this way, we have only two alignments for each dataset. Anyway, we remind that, as reported in the text, we check the alignment at each laser change by maximizing its transmittance signal.*

4) The $MAC_{BrC}$ reported in this study (9.8 $m^2g^{-1}$ @635 nm and 23 $m^2g^{-1}$ @405 nm) seems to be one magnitude higher than the literatures values. An example is shown below. The following table was adopted from (Updyke et al., 2012). The author argued the difference is due to the operative defined BrC mass used in this study. It should be noted that literature studies applied different technical approaches for $MAC_{BrC}$ determination as well, but most studies reported a $MAC_{BrC}<1$ $m^2g^{-1}$. The author should explain why their results are significantly different from previous studies.

Values of MAC in $m^2 g^{-1}$ from representative field and laboratory studies of organic aerosols measured at or near 500 nm. The cited MAC values explicitly remove contributions from black carbon. With the exception of this work, all data cited in this table likely correspond to primary sources of brown carbon.

| Sample (campaign) | λ (nm) | MAC ($m^2 g^{-1}$) | Reference |
|---|---|---|---|
| Brown carbon produced by aging SOA with 100 ppb $NH_3$ (lab) | 500 | 0.001−0.1 | This work |
| "Tar balls" from smoldering combustion of wood; brown carbon contribution (lab) | 532 | 0.01−0.07 (calculated from $k = 0.0005−0.003$) | (Chakrabarty et al., 2010) |
| HUmic-LIke Substances (HULIS) extracted from filter samples from various sites in Europe | 532 | 0.07−1 (calculated from $k = 0.003−0.05$) | (Dinar et al., 2008) |
| Methanol extracts from wood combustion particles (lab) | 500 | 0.1−0.5 (estimated from the graphs) | (Chen and Bond, 2010) |
| Refractory organic carbon from biomass burning in North America (INTEX/ICARTT) | 470 | 0.6 | (Clarke et al., 2007) |
| | 530 | 0.1 | |
| "Tar balls" in North America (YACS) | 632 | 0.4 (calculated from reported $k = 0.02$) | (Hand et al., 2005) |
| Brown carbon in pollution plumes from Asia (CAPMEX) | 532 | 0−1 | (Flowers et al., 2010) |
| Brown carbon in particles collected in Asia (EAST-AIRE) | 520 | 0.6 | (Yang et al., 2009) |
| Acetone extracts form biomass burning aerosols in Africa (SAFARI 2010) | 500 | 0.9 | (Kirchstetter et al., 2004) |
| Amorphous carbon spheres from biomass burning (ACE Asia) | 550 | 4 | (Alexander et al., 2008) |

*All MAC values in the table refer the optical "effect" of BrC compared to the OC from wood burning. In this sense, they are not considering the BrC molecules only but their optical response normalized to all the $OC_{WB}$ mass. Also in our case, if we plot $b_{abs(BrC)}$ vs. the whole $OC_{WB}$, we get $MAC(OC_{WB}) \approx (0.4 \pm 0.1)$ $m^2 g^{-1}$ for the red and $\approx (2.3 \pm 0.2)$ $m^2 g^{-1}$ for the blue (see also the last part of the §5 of the Massabò et al 2016 paper).*

5) Line 277. Regarding the BrC mass determination using the method reported in Massabò et al. (2016), did the author considered laser-temperature correction (Jung et al., 2011)? Seen from Fig 5 in Massabò et al. (2016), the laser signal keep increasing during the $CH_4$ stage, implying that laser-temperature correction was likely not performed. If that's case, the BrC mass should be re-calculated.

*We thank the Referee for the suggestion. However, we are aware of the reported issue and we actually applied an operative correction similar to what proposed in the quoted paper. So, we think that no further corrections are needed in our dataset. Anyway, the Referee is rising an important issue often neglected in TOT/TOR analysis (the effect of temperature on transmittance signal affecting EC/OC separation).*

6) In addition, even if the laser-temperature correction is accounted, the laser uncertainty is simply too high for BrC mass determination. Please specify the limit of quantification (LOQ) for OC in the OC/EC analysis. The reviewer feels that $LOQ_{OC}$ would be likely very close to the level of BrC reported in this study ($0.005 - 0.14$ µgC m$^{-3}$). If so, BrC reported using this approach is overinterpretation of the data.

*The limit of quantification of OC (LOQ) is 0.1 µgC cm$^{-2}$ that corresponds, in our sampling conditions, to about 0.009 µgC m$^{-3}$. So, except for few points, most of the BrC levels reported in the paper are well over this LOQ.*

7) The BrC determination approach described in Massabò et al. (2016) lacks physical meanings. The OC/EC split by the laser signal in the thermal optical analysis depends on two assumptions: (i) pyrolyzed organic carbon evolved before native EC during the oxygen stage. (ii) pyrolyzed organic carbon and native EC have the same MAC. However, both of these assumptions had been proved invalid (Yang and Yu, 2002; Yu et al., 2002; Subramanian et al., 2006). The approach that author used is a paradox: On one hand the authors report a $MAC_{BrC}$ that is larger than $MAC_{EC}$. On the other hand, the laser correction process itself is based on the assumption that $MAC_{BrC}=MAC_{EC}=MAC_{POC}$. In that sense, the carbon fraction corresponding to the different laser split time cannot be considered as BrC mass.

*The paper Massabò et al. 2016 has been already evaluated and revised in a full peer-review process. Even if we think that the revision should be limited to this paper and not to others in the literature, we appreciate the interest of the Referee and we would like to reply at his/her specific comments. Furthermore, the referee's criticism directly address the basic assumptions of the TOT analysis since the first papers by Birch and Cary, 1996. We have to note however, that assumption (ii) makes not necessary the (i) one. From his point of view just the $MAC_{EC}=MAC_{POC}$ assumption is part of the game.*

*A considerable amount of works in the literature deals about the assumptions at the basis of the TOT/TOR technique, mostly pointing out that often they are critical or unacceptable (and we were aware of them, see §6 of that paper). But, in the absence of a definitive solution of these criticisms, these assumptions are necessary to separate EC from OC, or we simply have to forget this analytical technique.*

*The previous paper (Massabò, 2016) started from these (questionable, we agree) assumptions, trying to gather information on BrC, and actually we don't see any paradox. About the reported $MAC_{BrC}$, this value turned out from the comparison of: 1) difference in mass between uncorrected/corrected EC values and 2) absorption coefficients apportioned with the MWAA model to BrC. About the "other hand assumption", we did not write in that text the statement $MAC_{BrC}=MAC_{EC}=MAC_{POC}$, and this assumption is never considered in that proposed methodology. Anyway, we underline that we defined the retrieved BrC mass as "operative BrC*

*mass", precisely to highlight its dependence on the specific methodology adopted and, from this quantity, we calculated the $MAC_{BrC}$ (without the cited assumption $MAC_{BrC}=MAC_{EC}=MAC_{POC}$).*

8) The authors are encouraged to check the $b_{abs,BCff}$ vs. levo scatter plot. If the $R^2(b_{abs,BCff}$ vs. levo) is significantly lower than the $R^2(b_{abs,BrC}$ vs. levo), that would be a useful evidence to confirm a successful split of $b_{abs}$ into BrC, $BC_{WB}$ and $BC_{ff}$.

*As suggested, we checked the $b_{abs,BCff}$ vs. levo scatter plot and, as expected, there is no correlation between the two parameters. In the following, we report the two scatter plots.*

[Figure]

[Figure]

**Technical comments:**

1) The figure quality needs to be improved. For example, for comparison of the same quantity/parameter, the X and Y range should be the same and the aspect ratio of the plot should be 1:1.
   *In the final version of the paper, a full (graphical) revision of the figures will be given.*

2) Figure 1. Please label the laser wavelength on the photo directly for easy reference.
   *Done*

3) Figure 2-7. The font size is too small for the text in these figures. Please adjust accordingly.
   *Done*

4) Figure 5 caption. "WW and FF stand for Fossil Fuel and Wood Burning, respectively." Should be "FF and WB stand for Fossil Fuel and Wood Burning, respectively"
   *Done*

5) Line 245. "and biomass burning (WB)" should be wood burning?
   *Done*

**References**

Chen, L. W. A., Chow, J. C., Wang, X. L., Robles, J. A., Sumlin, B. J., Lowenthal, D. H., Zimmermann, R., and Watson, J. G.: Multi-wavelength optical measurement to enhance thermal/optical analysis for carbonaceous aerosol, Atmos. Meas. Tech., 8, 451-461, doi: 10.5194/amt-8-451-2015, 2015.

Chow, J. C., Wang, X., Sumlin, B. J., Gronstal, S. B., Chen, L. W. A., Trimble, D. L., Kohl, S. D., Mayorga, S. R., Riggio, G., Hurbain, P. R., Johnson, M., Zimmermann, R., and Watson, J. G.: Optical Calibration and Equivalence of a Multiwavelength Thermal/Optical Carbon Analyzer, Aerosol. Air. Qual. Res., 15, 1145-1159, doi: 10.4209/aaqr.2015.02.0106, 2015.

Chow, J. C., Watson, J. G., Green, M. C., Wang, X., Chen, L. W. A., Trimble, D. L., Cropper, P. M., Kohl, S. D., and Gronstal, S. B.: Separation of brown carbon from black carbon for IMPROVE and Chemical Speciation Network PM2.5 samples, J. Air Waste Manage. Assoc., 68, 494-510, doi: 10.1080/10962247.2018.1426653, 2018.

Hadley, O. L., Corrigan, C. E., and Kirchstetter, T. W.: Modified Thermal-Optical Analysis Using Spectral Absorption Selectivity To Distinguish Black Carbon from Pyrolized Organic Carbon, Environ. Sci. Technol., 42, 8459-8464, doi: 10.1021/Es800448n, 2008.

Jung, J., Kim, Y. J., Lee, K. Y., Kawamura, K., Hu, M., and Kondo, Y.: The effects of accumulated refractory particles and the peak inert mode temperature on semi-continuous organic carbon and elemental carbon measurements during the CAREBeijing 2006 campaign, Atmos. Environ., 45, 71927200, doi: 10.1016/j.atmosenv.2011.09.003, 2011.

Massabò, D., Caponi, L., Bove, M. C., and Prati, P.: Brown carbon and thermal–optical analysis: A correction based on optical multi-wavelength apportionment of atmospheric aerosols, Atmos. Environ., 125, 119-125, doi: 10.1016/j.atmosenv.2015.11.011, 2016.

Subramanian, R., Khlystov, A. Y., and Robinson, A. L.: Effect of peak inert-mode temperature on elemental carbon measured using thermal-optical analysis, Aerosol. Sci. Technol., 40, 763-780, doi: 10.1080/02786820600714403, 2006.

Updyke, K. M., Nguyen, T. B., and Nizkorodov, S. A.: Formation of brown carbon via reactions of ammonia with secondary organic aerosols from biogenic and anthropogenic precursors, Atmos. Environ., 63, 22-31, doi: 10.1016/j.atmosenv.2012.09.012, 2012.

Yang, H. and Yu, J. Z.: Uncertainties in charring correction in the analysis of elemental and organic carbon in atmospheric particles by thermal/optical methods, Environ. Sci. Technol., 36, 5199-5204, doi: 10.1021/Es025672z, 2002.

Yu, J. Z., Xu, J. H., and Yang, H.: Charring characteristics of atmospheric organic particulate matter in thermal analysis, Environ. Sci. Technol., 36, 754-761, doi: 10.1021/Es015540q, 2002.

---

## Author Comment (AC3) · 17 Apr 2019

**Review report AMT 2019-5 manuscript version 1**

*We strongly acknowledge the Referee for the valuable and precious comments and suggestions.*

The paper describes the application of a modified Sunset Lab Inc. EC/OC analyzer with a two wavelength set-up for analysis of ambient aerosol samples. It strongly relates to earlier work of the author which is described detailed in previous publications, nevertheless it extends to additional findings. The results derived from the comparison of two temperature protocols, NIOSH5040 and EUSAAR2, for the 405 nm wavelength as well as the reported MAC values are a valuable addition to the literature. However, the paper lacks in structure and suffers in use of proper scientific English which limits its potential. It is recommended that the text is reviewed with focus on syntax and vocabulary. Particular attention should be given in the first paragraph of the abstract and the first two paragraphs of the introduction. Further:

*We tried to improve English for both syntax and vocabulary. In the following, a point-by-point reply to all the comments is given.*

Sentence starting in line 115: "The hypothesis under such choice…" should be rephrased.

*Done*

The terms "real samples", "real-world samples" and "real-world aerosol samples" are used throughout the text. The use of one term is recommended.

*We changed using the form "real-world aerosol samples throughout the text.*

"λ=" , "@λ"  and "@λ=" are used to state wavelengths. Please consider using one form (λ=) for consistency.

*Done*

Sentence starting in line 59 should be revised. "Standarized" should be replaced by standardized. Since 2017, when EN16909 was published, there is uniformity in OC/EC analysis methodology, at least for EU.

*Done*

Paragraph starting in line 157: Any specific reason why this subset was analyzed with EUSAAR2 only?

*After the first positive results with pure Aquadag samples and considering the slope close to one, we preferred to make a "reproducibility study" (i.e. we used 2 punches for each laser in each sample) by EUSAAR_2 protocol. We got a very good agreement and final results only have been reported in the text. We'll make clear this point in the revised text.*

Paragraph starting in line 165: PM10 samples are known to add complexity in OC/EC analysis due to minerals, refractory material and oxides present in coarse fraction. Did you consider sampling/analysis of PM2.5 samples and have you noticed any of the above interferences?

*This point is very interesting, we thank the Referee for this valuable suggestion. Unfortunately, we had not the possibility to apply our methodology on PM2.5 samples but for sure we will try in the next future.*

Line 174: It sounds like two different subsets were created, one for analysis with EUSAAR2 and one for NIOSH5040. If that is the case, why was that choice made instead of all samples being analyzed with both protocols?

*As explained above, we preferred to have a reproducibility check by measuring 2 punches for each laser in each sample.*

Line 193: It is not clear to me why the discrepancy between EUSAAR2 and NIOSH5040 is mainly driven by charring. In a sense more pyrolytic carbon would result in a later split point and less EC reported. Further, since the blue laser diode resulted in later split points for EUSAAR2, wouldn't that rate it as less sensitive to charring instead of more, as mentioned in the text?

*Actually, as shown in the example below (please note that the red and blue transmittance are normalized at the same initial value), in the EUSAAR_2 thermograms we observe a steeper decrease of the "blue" transmittance in the first phases (low temperatures). This corresponds to a higher sensitivity to pyrolytic formation when we use the blue laser diode.*

[Figure]

*EUSAAR_2*

[Figure]

*NIOSH5040*

Is it possible to include a figure and/or representative thermograms that illustrate the consistent 40% discrepancy between EUSAAAR2 and NIOSH5040?

*As reported in the text we unfortunately had not the possibility to repeat the NIOSH and EUSAAR_2 protocols at the two different wavelengths on the same filters. This is why we based our discussion on the analysis of literature and previous results. We anyway include here two typical thermograms, NIOSH and EUSAAR_2, of the same urban samples batch and showing the typical transmittance trend linked to the quoted discrepancy.*

[Figure]

*Eusaar2, split point ≈ 950 s*

[Figure]

*Niosh5040, split point ≈ 480 s*

Line 267 and elsewhere in the text: The term "Sunset set-ups" could be altered to a more descriptive term.

*Done*

Line 289: What would be the value of 1σ? It seems that the difference between the two MAC values reported is substantially greater than the reported uncertainty.

*The Referee is right, we did a material mistake in the text: the one sigma uncertainty is 0.4 m² g⁻¹ (and this is right) but the discrepancy between the MAC values calculated in 2016 and the present work is slightly above 3 sigma (we change the text accordingly).*

Line 316: "Brow carbon"

*Done*

Line 338: This sentence could be rephrased for easier comprehension.

*We rephrased the sentence in this way: "We retrieved Brown Carbon concentration values directly from the Sunset thermograms following Massabò et al., 2016. Exploiting the synergic information provided by the Multi Wavelength Absorbance Analyzer, MWAA (Massabò et al., 2015) we could obtain the MAC(BrC) at the two wavelengths".*

Line 348: Same stands for this sentence.

*We rephrased the sentences in this way: "In our findings, the ratio between BrC and Levo concentration values depends on the wavelength of the transmittance signal adopted during the thermo-optical analysis. This behavior could be due to 1) a better accuracy of the results in blue-light, more sensitive to BrC, or 2) the definition of BrC itself, which has to be considered wavelength-dependent. The present results do not allow any conclusive statement on this issue: actually, the label "Brown Carbon", as well as the widely used "Organic and Elemental Carbon", comes from an operative definition not without ambiguity.*

Figure 4: It is not clear which relationship applies to which trendline.

*Figure 4 has been amended.*

Figure 6: It seems that 2 separate subgroups are formed, one equal and above the trendline and one below the trendline. Are those related to the specific sampling strategy or to any other parameter?

*We noted this strange behavior with the formation of two separate subgroups. It doesn't depend on different sampling strategy neither other evident parameter. Unfortunately, we didn't find a reasonable explanation for this.*

---

## Author Response (AR3)

Done

Done

Associate Editor Decision: Publish subject to minor revisions (review by editor) (09 May 2019) by Willy Maenhaut

Comments to the Author:

The authors have reasonably addressed the comments of the three anonymous referees and they have modified their manuscript accordingly. However, the comments below should be taken into consideration before the manuscript can be published in AMT.

We changed the text following all the requested modifications. We thank the Editor for the accurate work and all the suggestions. We have also improved the figures (removing useless information too) and corrected the number of the last paragraph.

Line 1 and further throughout the manuscript: replace "thermo-optical" by "thermal-optical".

Lines 23 and 169: replace "Genoa (IT)" by "Genoa (Italy)".

Lines 25, 27 and 251: replace "at both the" by "at both".

Lines 43, 94 and 170: replace "e.g. " by "e.g., ".

Line 43: replace "reference therein" by "references therein".

Line 46: replace "Brown Carbon" by "Brown Carbon (BrC)" so that BrC is defined for later use within the main text.

Line 55 and on 8 other occasions further within the text: replace "i.e. " by "i.e., ".

Line 62: replace "of TC, OC and EC" by "of thermal, organic and elemental carbon (TC, OC and EC)" so that TC, OC and EC are defined for later use within the main text.

Lines 70 and 114: replace "well know" by "well-known".

Line 76: replace "to not say" by "not to say".

Line 77: abbreviations and acronyms, here "TOT/TOR" should be defined (written full-out) when first used; thus, replace "TOT/TOR instrument" by "thermal-optical reflectance / thermal-optical transmittance (TOT/TOR) instrument".

Line 84: replace "7-wavelengths" by "7 wavelengths".

Line 84 and further throughout the manuscript: replace "thermal/optical" by "thermal-optical".

Line 86: replace "can be also" by "can also be".

Line 122: replace "smaller of the" by "smaller than the".

Line 148: replace "composed by" by "composed of".

Line 149: replace "MWAA instrument (Multi Wavelength Absorbance Analyzer," by "MWAA instrument (";
MWAA should not be defined again as it was already defined in lines 81-82.

Line 154: replace "the two Sunset" by "the new blue-light and original Sunset".

Line 155: replace "resulted compatible adopting" by "were in excellent agreement for".

Line 161: replace "resulted independent" by "was independent".

Line 171: replace "sizeable contributes of" by "a sizeable contribution from".

Line 182: replace "resulted lower by about 30%" by "resulted in about 30% lower values".

Line 186: replace "e.g.:" by "e.g.,".

Lines 196-197: "and thus possibly more reliable in the EC/OC separation" should be rephrased; a verb is missing here and it is unclear to me what the authors want to say.

Line 204: replace "collected wintertime" by "collected during wintertime".

Line 219: replace "All the samples" by "All filters".

Line 230: replace "well known" by "well-known" and replace "typical marker" by "typical markers".

Line 245: replace "405 nm" by "405 nm,".

Line 249: replace "contribute to" by "contribution to".

Line 263: replace "i.e.:" by "i.e.," and replace "Sunset unit" by "a Sunset unit".

Line 292: replace "obtained in the" by "obtained for the".

Line 320: replace "of the more" by "than is the case for the more".

Line 324: replace "doubles the" by "is twice the".

Line 339: replace "collected mostly" by "collected during mostly".

Line 344: replace "performed in the" by "performed for the".

Line 345: replace "At our" by "To our".

Line 353: replace "definition not" by "definition which is not".

Lines 390-394: "Chen, Y. and Bond, 2010" should come after "Chen, L. W. A. et al., 2015".

Lines 458-462: "Lack, D. A. et al., 2012" should come after "Lack, D. A. and Langridge, 2013".

Line 540: replace "and the comparison" by "and comparison".

Line 555: replace "experimental AAE values" by "experimental AAE (AAEexp) values".

Line 579, Figure 3: within the figure, to the right of the equation in the bottom right corner, replace "405nm" by "635nm".